# mTOR-Dependent Role of Sestrin2 in Regulating Tumor Progression of Human Endometrial Cancer

**DOI:** 10.3390/cancers12092515

**Published:** 2020-09-04

**Authors:** Jiha Shin, Jeongyun Bae, Sumi Park, Hyun-Goo Kang, Seong Min Shin, Gunho Won, Jong-Seok Kim, Ssang-Goo Cho, Youngsok Choi, Sang-Muk Oh, Jongdae Shin, Jeong Sig Kim, Hwan-Woo Park

**Affiliations:** 1Department of Cell Biology, Konyang University College of Medicine, Daejeon 35365, Korea; dixest0337@gmail.com (J.S.); 0518bjy@gmail.com (J.B.); psm0503@konyang.ac.kr (S.P.); hyungoo89@naver.com (H.-G.K.); ssminhk@gmail.com (S.M.S.); shinjd@konyang.ac.kr (J.S.); 2Myunggok Medical Research Institute, Konyang University College of Medicine, Daejeon 35365, Korea; wgh007@hanmail.net (G.W.); jskim7488@konyang.ac.kr (J.-S.K.); sangmuk_oh@konyang.ac.kr (S.-M.O.); 3Department Centers for Disease Control & Prevention, National Institute of Health, Cheongju 28159, Korea; 4Department of Stem Cell and Regenerative Biotechnology, Konkuk University, Seoul 05029, Korea; ssangoo@konkuk.ac.kr (S.-G.C.); choiys911@gmail.com (Y.C.); 5Department of Biochemistry, Konyang University College of Medicine, Daejeon 35365, Korea; 6Department of Obstetrics and Gynecology, Soonchunhyang University Seoul Hospital, Seoul 04401, Korea

**Keywords:** endometrial cancer, sestrin2, mTORC1, proliferation, migration, reactive oxygen species (ROS)

## Abstract

**Simple Summary:**

Mammalian target of rapamycin complex 1 (mTORC1), a key controller of growth and environmental stress signaling, is frequently activated in human cancers. Sestrin2 (SESN2), a highly conserved stress-inducible protein, is one of the negative feedback mechanisms for inhibiting chronic activation of mTORC1. This study aimed to investigate the expression and clinical implications of SESN2 in endometrial cancer using an in vitro and in vivo approach. The analysis indicated increased levels of SESN2 and mTORC1 pathway activity in cancer tissues than in normal tissues. High SESN2 expression correlated with shorter patient survival duration. However, lentiviral overexpression of *SESN2* and mTOR inhibitors suppressed cancer cell proliferation, migration, and epithelial–mesenchymal transition. Our study provides strong evidence for prognostic significance of SESN2, and its association with mTORC1 pathway and endometrial cancer growth. Thus, the results identified SESN2 as a potential therapeutic target in endometrial cancer.

**Abstract:**

Oncogenic activation of the mammalian target of rapamycin complex 1 (mTORC1) leads to endometrial cancer cell growth and proliferation. Sestrin2 (SESN2), a highly conserved stress-inducible protein, is involved in homeostatic regulation via inhibition of reactive oxygen species (ROS) and mTORC1. However, the role of SESN2 in human endometrial cancer remains to be investigated. Here, we investigated expression, clinical significance, and underlying mechanisms of SESN2 in endometrial cancer. SESN2 was upregulated more in endometrial cancer tissues than in normal endometrial tissues. Furthermore, upregulation of SESN2 statistically correlated with shorter overall survival and disease-free survival in patients with endometrial cancer. SESN2 expression strongly correlated with mTORC1 activity, suggesting its impact on prognosis in endometrial cancer. Additionally, knockdown of *SESN2* promoted cell proliferation, migration, and ROS production in endometrial cancer cell lines HEC-1A and Ishikawa. Treatment of these cells with mTOR inhibitors reversed endometrial cancer cell proliferation, migration, and epithelial–mesenchymal transition (EMT) marker expression. Moreover, in a xenograft nude mice model, endometrial cancer growth increased by *SESN2* knockdown. Thus, our study provides evidence for the prognostic significance of SESN2, and a relationship between SESN2, the mTORC1 pathway, and endometrial cancer growth, suggesting SESN2 as a potential therapeutic target in endometrial cancer.

## 1. Introduction

Endometrial cancer is the most common malignant cancer of the female reproductive system, with an increasing incidence worldwide due to high risk factors, including increased obesity rates [1,2,3]. Patients with endometrial cancer diagnosed at an early stage show good prognosis with high survival rates, while those diagnosed at late stage have limited treatment options and poor prognosis, with low survival rates [4,5]. The poor prognosis could be an outcome of progression in endometrial cancer, including invasion of myometrium and lymph node metastasis that could be affected by reactive oxygen species (ROS) or hypoxia [6,7]. Better understanding of the underlying molecular mechanisms of endometrial carcinogenesis may improve diagnosis and provide insights for development of effective treatment strategies.

The mammalian target of rapamycin (mTOR), a serine/threonine kinase that regulates cell growth and metabolism, is frequently activated in human cancers [8,9]. It comprises two structurally and functionally distinct complexes—mTOR complex 1 (mTORC1) and mTORC2. The mTORC1 is activated by nutrients via the phosphoinositide 3-kinase (PI3K)/AKT pathway, which is the most common cancer-promoting signaling event [10,11]. Hyperactivation of mTORC1 directly activates p70 ribosomal protein S6 kinase (p70S6K) and inhibits eIF4E binding protein (4EBP) to increase protein synthesis, which can cause endoplasmic reticulum (ER) stress and tumor progression [12,13]. However, inhibition of the mTORC1 signaling pathway with an immunosuppressive drug, rapamycin, suggested mTORC1 to be a potential target for anti-cancer treatment [14,15]; although, molecular mechanisms of mTORC1 regulation in endometrial cancer cells remain poorly understood.

Sestrins (SESNs) are a highly conserved protein family comprising SESN1, SESN2, and SESN3 in mammals, and can be induced by various cellular stresses including DNA damage, increased ROS, hypoxia, and ER stress [12,16,17,18]. SESN2 maintains metabolic homeostasis and prevents age- and obesity-related pathologies by inhibiting ROS accumulation and the mTORC1 pathway [12,18,19,20]. *SESN2*, which is homologous to the *SESN1*, is induced by genotoxic and oxidative stresses in a p53-dependent manner [19,21], while SESN2 induction in response to hypoxia and ER stress is p53-independent [12,22]. Chronic activation of mTORC1 is associated with increased risk of various metabolic diseases, and the transcriptional activation of SESN2 is one of the negative feedback mechanisms for inhibiting chronic activation of mTORC1 [12,23,24]. Upon induction, SESN2 inhibits the activation of mTORC1 kinase in an AMP-activated protein kinase (AMPK) and tuberous sclerosis complex (TSC)- or GAP activity toward Rags (GATOR)-dependent manner [19,25,26]. Although multiple studies have documented the role of SESN2 in hepatocellular carcinoma, non-small cell lung cancer, bladder cancer, and colon cancer [27,28,29,30], there are conflicting results regarding its expression in hepatocellular carcinoma [28,31]. However, clinical significance of SESN2 in endometrial cancer, and its role in the regulation of endometrial cancer cell proliferation and migration, remains unexplored.

In this study, for the first time, we examined SESN2 expression levels in the primary tumors of endometrial cancer patients and in the endometrial cancer samples of The Cancer Genome Atlas (TCGA), and their correlations with clinicopathological factors. We also investigated the role of SESN2 expression in endometrial cancer and the molecular mechanisms by which SESN2 regulates cancer cell growth and migration. Our results suggested SESN2 expression to be significantly upregulated in endometrial cancer tissues, correlating with an increased mTORC1 pathway. The Kaplan–Meier analysis identified SESN2 as a putative prognostic factor in endometrial cancer. Moreover, knockdown of *SESN2* promoted endometrial cancer cell proliferation, migration, and ROS production via the mTORC1-dependent pathway. We also observed that knockdown of *SESN2* enhanced tumor growth in endometrial cancer cells implanted in nude mice. Thus, our study implicates SESN2 to be a potential candidate in the treatment of endometrial cancer.

## 2. Results

### 2.1. SESN2 Expression and Its Clinical Significance in Endometrial Cancer

We evaluated the mRNA expression of *SESN2* in the surgical endometrial cancer tissue samples and normal endometrium samples using quantitative real-time polymerase chain reaction (qRT-PCR). *SESN2* mRNA levels are significantly more elevated in endometrial cancer tissues than that in normal endometrial tissues (Figure 1A). Furthermore, we tested the protein expression of SESN2 using immunoblotting in the endometrial cancer and normal tissues. Consistent with the mRNA expression, immunoblot data showed SESN2 levels to be significantly more increased in endometrial cancer tissues than that in normal endometrial tissues (Figure 1B). Next, to investigate the prognostic significance of SESN2 in endometrial cancer, we examined its expression in cancer and corresponding normal counterparts using TCGA database. The *SESN2* mRNA levels were significantly more increased in the tumor than in normal tissues in TCGA dataset (*p* < 0.05) (Figure 1C). Additionally, immunohistochemistry staining results validated from the Human Protein Atlas database revealed the SESN2 protein to be downregulated in normal tissues and upregulated in endometrial cancer tissues (Figure 1D). Further, we performed Kaplan–Meier survival analyses to investigate the correlation of SESN2 expression with overall survival and disease-free survival in endometrial cancer patients. Results showed that high SESN2 expression was associated with significantly decreased overall survival (*p* = 0.018) and disease-free survival (*p* = 0.032) in patients with endometrial cancer (Figure 1E,F). Taken together, these results suggest that SESN2 expression affects the prognosis in endometrial cancer.

### 2.2. Expression of SESN2 Correlates with mTOR Pathway Activity in Endometrial Cancer

Deregulation of the mTOR pathway is implicated in pathogenesis of several cancer types [32]. Constitutive activation of the mTORC1, as one of the mTOR complexes, is identified in a few human cancers [33]. Therefore, we analyzed the levels of downstream target proteins of mTORC1 in endometrial cancer and normal tissues using immunoblotting. The analysis indicated phosphorylation of ribosomal protein S6, which is mediated by an mTORC1 target p70S6K, to be significantly more elevated in endometrial cancer tissues than that in normal tissues (Figure 2A). mTORC1 is composed of mTOR, raptor (regulatory associated protein of mTOR), mLST8, PRAS40, and deptor (DEP domain containing mTOR interacting protein). Rheb GTPase directly binds to mTOR and activates mTORC1 in response to growth factor signals [34]. Further, to investigate this association, we evaluated the correlation between *SESN2* and mTOR pathway-related markers, including *RPTOR*, *MTOR,* and *RHEB,* by analyzing RNA expression data from cancer patients using the Gene Expression Profiling Interactive Analysis (GEPIA) database. The results revealed a significant positive correlation between the expression of *SESN2* and *RPTOR* in endometrial cancer patient tissues (r = 0.34, *p* = 4.5 × 10^−6^) (Figure 2B). However, no significant correlations were observed between *SESN2* and *MTOR* or *RHEB* (*p* > 0.05) (Figure 2C,D). Reasons for the correlations between *SESN2* and *MTOR* or *RHEB* being not significant could be because mRNA levels of the core components of mTORC1 do not reflect mTORC1 kinase activity. These results suggest that mRNA levels of *SESN2* positively correlate with those of few mTOR signaling pathway genes in endometrial cancer.

Protein levels of SESN2 are induced upon chronic activation of mTORC1, which subsequently inhibit mTORC1 activation [12,20,24]. To evaluate the regulatory effect of SESN2 on the mTORC1 pathway, lentiviral vectors expressing short hairpin RNAs (shRNAs) against human *SESN2* or control were transduced into HEC-1A and Ishikawa cells. After transduction, expression of SESN2 significantly reduced in the HEC-1A and the Ishikawa cells (Figure 2E). Moreover, the knockdown of *SESN2* enhanced phosphorylation of S6 and p70S6K proteins (Figure 2F), indicating SESN2 expression to negatively regulate mTORC1 activity in endometrial cancer.

### 2.3. Knockdown of SESN2 Promotes Endometrial Cancer Cell Proliferation and ROS Production

To determine the effect of SESN2 in endometrial cancer cell proliferation, a water-soluble tetrazolium salt (WST-1) assay was performed every day for 1–5 days post-transduction. The results showed that knockdown of *SESN2* significantly increased cell proliferation in HEC-1A and Ishikawa cells (Figure 3A). We also examined the expression of proliferation marker gene *MKI67* in cells with depleted *SESN2* expression in HEC-1A and Ishikawa cells using qRT-PCR. The results suggested more elevated mRNA levels of *MKI67* upon knockdown of *SESN2* than those in control cells (Figure 3B). Next, we investigated the correlation between *SESN2* and cell cycle-associated genes, including *CDKN1A* (encoding p21) and *CDKN1B* (encoding p27), by analyzing RNA expression data for cancer patients using the GEPIA database. The analysis indicated a significant positive correlation of expression of *SESN2* with *CDKN1A* (r = 0.3, *p* = 6.5 × 10^−5^) and *CDKN1B* (r = 0.18, *p* = 0.016) (Figure 3C,D). Moreover, consistent with the correlation analyses, qRT-PCR analysis showed significantly lower expression of CDKN1A and CDKN1B in cells with depleted *SESN2* levels than those in control (Figure 3E). These findings suggest that downregulation of *SESN2* facilitates proliferation and cell cycle progression in endometrial cancer cells.

Furthermore, the endogenous levels of ROS can regulate several cellular physiological processes, including cell proliferation and survival [35]. However, ROS overproduction is implicated in the pathogenesis of cancer and resistance to treatment [36,37]. SESN2 is known to inhibit ROS through their enzymatic activity or by nuclear factor erythroid 2-related factor 2 (Nrf2) activation [21,38]. Thus, to investigate whether SESN2 is involved in ROS generation in HEC-1A and Ishikawa cells, the fluorescence probe, CM-H_2_DCFDA (5-(and-6)-chloromethyl-2′, 7′-dichlo-rodihydrofluorescein diacetate, acetyl ester), was used to analyze the intracellular ROS levels. The analysis suggested significantly more increased ROS levels in *SESN2* knockdown cells than those in control cells (Figure 3F).

### 2.4. SESN2 Regulates EMT and Migration in Endometrial Cancer Cells

The epithelial–mesenchymal transition (EMT) is important for the initiation of cancer cell dissemination and metastasis [39]. Thus, we investigated whether expression of *SESN2* correlates with that of the EMT markers, including *CDH1* (encoding E-cadherin) and *CDH2* (encoding N-cadherin), in the same dataset of endometrial cancer patients. The analysis indicated expression of CDH1 to be highly correlated with that of *SESN2* (r = 0.34, *p* = 5.3 × 10^−6^) (Figure 4A). However, no significant correlation was observed with *CDH2* (*p* > 0.05) (Figure 4B). Next, we studied the role of SESN2 in modulating expression of these EMT markers using cell-based assays. Immunoblot analysis suggested knockdown of *SESN2* to decrease and increase the protein levels of E-cadherin and N-cadherin, respectively (Figure 4C). Further, the function of SESN2 in endometrial cancer cell migration was evaluated using the wound healing scratch assay. The results suggested more significant promotion of cell migration in HEC-1A and Ishikawa cells with knockdown of *SESN2* than that with control (Figure 4D,E).

### 2.5. SESN2 Regulates Endometrial Cancer Cell Proliferation and ROS Production in a mTOR-Dependent Manner

As SESN2 inhibits the mTOR activity, we checked whether mTORC1 inhibition in cells with depleted levels of *SESN2* influences endometrial cancer cell proliferation and ROS production. We treated *SESN2* knocked-down HEC-1A and Ishikawa cells with mTOR inhibitors, rapamycin or Torin 1, and performed immunoblot analysis. The results suggested that rapamycin and Torin 1 more effectively suppressed phosphorylation of S6 and p70S6K in cells with depleted *SESN2* levels than that in control cells (Figure 5A). Moreover, treatment of rapamycin and Torin 1 more significantly inhibited cell proliferation in *SESN2* knocked-down cells than that in control cells (Figure 5B). The qRT-PCR analysis showed a more significant decrease in mRNA levels of *MKI67* in rapamycin- or Torin 1-treated cells than that in control cells (Figure 5C). Further, to investigate whether the effect of SESN2 on ROS production was dependent on the mTORC1 pathway in endometrial cancer cells, we measured ROS levels using the CM-H_2_DCFDA probe in HEC-1A and Ishikawa cells treated with rapamycin and Torin 1. The analysis suggested a reduction in oxidative stress in *SESN2* knocked-down cells treated with rapamycin and Torin 1 (Figure 5D). Thus, these results indicate that SESN2 attenuates endometrial cancer cell proliferation and ROS production by suppressing activity of mTORC1.

### 2.6. SESN2 Regulates EMT and Migration in Endometrial Cancer Cells via the mTORC1 Pathway

To further explore the effect of mTORC1-inhibition on endometrial cancer cell migration, we performed a wound healing scratch assay in HEC-1A and Ishikawa cells with depleted levels of *luciferase* and *SESN2*. The results showed that treatment of rapamycin and Torin 1 significantly suppressed migration of cells with depleted *SESN2,* more than that in control cells (Figure 6A,B). Moreover, Torin 1 treatment also inhibited migration of control cells, more than that of *SESN2*-depleted cells (Figure 6A,B).

Next, we evaluated the changes in the expression of E-cadherin and N-cadherin in response to mTOR inhibition. Immunoblot analysis showed that treatment with rapamycin and Torin 1 or *SESN2* overexpression increased E-cadherin expression and reduced N-cadherin expression in *SESN2* knockdown cells (Figure 7A and Appendix A). Whereas, mTORC1 activation by *TSC2*-shRNA resulted in a decrease and increase in E-cadherin and N-cadherin protein levels, respectively (Figure 7B). Further, we investigated the effect of *SESN2* overexpression on E-cadherin and N-cadherin expression in the cells. Interestingly, cells overexpressing SESN2 showed lower levels of N-cadherin than those overexpressing control GFP (Figure 7C); but, E-cadherin protein levels remained unchanged in both cell groups, probably due to sufficient levels of SESN2 in these cells. Taken together, these results suggest that SESN2 suppresses EMT and migration in endometrial cancer cells via a mTORC1-dependent mechanism.

### 2.7. SESN2 Regulates Tumor Growth in a Xenograft Nude Mice Model

To further determine the role of SESN2 in the progression of endometrial cancer in vivo, nude mice were subcutaneously injected with control *luciferase* or *SESN2* knocked-down HEC-1A cells. The analysis suggested *SESN2*-depleted xenografts to be significantly larger in volume and weight (Figure 8A–C). Furthermore, qRT-PCR analysis indicated significantly more increased mRNA levels of *MKI67* in *SESN2*-depleted xenografts than those in control xenografts (Figure 8D). Next, we investigated the influence of SESN2 on expression of E-cadherin and N-cadherin in vivo. Immunoblot analysis revealed reduced and increased levels of E-cadherin and N-cadherin in *SESN2*-depleted xenografts (Figure 8E). Additionally, the analysis also confirmed the inhibitory effect of *SESN2* on mTORC1 in *SESN2*-depleted tumors (Figure 8E). Taken together, our results suggest that SESN2-mediated inhibition of mTORC1 might be one of the mechanisms responsible for therapeutic potential in endometrial cancer.

## 3. Discussion

Enhanced activity of the mTORC1 pathway is well documented in few human cancers [33], although limited studies discuss the regulation of the mTOR pathway in endometrial cancer [40,41]. Here, we observed more increased mTORC1 activity in endometrial cancer tissues than that in normal endometrial tissues, as evidenced by the elevated phosphorylation of S6 protein. Activation of mTORC1 regulates cell proliferation and survival, which increases ribosome biogenesis and translation, and represses autophagy [42]. Induction of SESN2 is a negative feedback mechanism that suppresses chronic activation of mTORC1 [12,23,24]. Furthermore, inhibition of mTORC1 activity is closely related to abrogation of tumor growth and progression [43]. However, the molecular relationship between SESN2 and the mTORC1 pathway in endometrial cancer remained to be understood. In the present study, we showed increased expression of SESN2 mRNA and protein in endometrial cancer tissues. The elevated expression of SESN2 can be correlated with regulation of mTORC1 signaling in endometrial cancer tissues (Figure 8F). Furthermore, the Kaplan–Meier survival analysis suggested that SESN2 could serve as a diagnostic marker for the development of endometrial cancer.

SESN2 is a highly conserved stress-inducible protein that can be increased in cells exposed to various stresses, including DNA damage, hypoxia, and oxidative stress [22,24,44]. However, numerous other studies indicated lower expression of SESN2 in several cancers, including hepatocellular carcinoma, non-small cell lung cancer, bladder cancer, and colon cancer tissues than that in non-cancerous tissues [27,28,29,30]. Decreased expression of SESN2 correlated with tumor progression [27,28]. Additionally, low levels of SESN2 are associated with poor prognosis in patients with different cancer types [28,29,45]. In contrast, our results clearly demonstrate upregulation of SESN2 mRNA and protein levels in endometrial cancer tissues. Similar changes are also reported in hepatocellular carcinoma tissues [31]. As ROS, hypoxia, inflammation, and ER stress are associated with cancer initiation and progression [46,47], it is plausible that these stresses contribute to the increased expression of SESN2 in endometrial cancer tissues.

High SESN2 expression was associated with poor prognosis in endometrial cancer patients, whereas knockdown of SESN2 promoted cell proliferation and migration in endometrial cancer cell lines HEC-1A and Ishikawa. Although SESN2 expression was correlated with the mTOR signaling pathway, SESN2 may not be the direct cause of increased mTORC1 activity, but rather the opposite—that increased mTORC1 activity caused the increase in SESN2. In endometrial cancer patient tissues, inhibition of mTORC1 activity was not effectively mediated by the negative feedback regulation of SESN2, whereas in endometrial cancer cell lines, lentiviral overexpression of *SESN2* effectively suppressed mTORC1 activity. A possible reason for the disagreement between patient and cell line data may be due to the bidirectional role of SESN2 in endometrial cancer progression. Activation of multiple stresses, including ER stress and oxidative stress, by dysregulated activation of mTORC1, initiates a negative-feedback loop to suppress mTORC1 activity by transcriptional activation of SESN2 during cancer cell growth and expansion [12,48,49]. We found that *SESN2* knockdown enhanced mTORC1 activation in endometrial cancer cell lines. Moreover, overexpression of *SESN2* reduced mTORC1 activation in endometrial cancer cell lines. However, mTORC1 activity was significantly higher in endometrial cancer tissues of the patients than in normal endometrial tissues, despite a significant increase in SESN2 expression. These findings may be the result of multiple molecular events that can lead to hyperactivation of mTORC1 during endometrial cell growth and expansion [48,50]. Thus, it appears that poor prognosis in endometrial cancer patients is caused by hyperactivation of the mTORC1 pathway rather than by high SESN2 expression.

The Cip/Kip family members, CDKN1A and CDKN1B, can arrest cell proliferation and were initially considered as tumor suppressors [51]. However, several studies have implicated their dual roles in cancer as tumor suppressor and oncogene [52,53,54]. Llanos et al. reported that CDKN1A expression is detected in proliferating head and neck squamous cell carcinoma cells and the growth inhibitory effect of CDKN1A is abrogated in these tumors [53]. We showed that knockdown of *SESN2* increased mTORC1 activity and cell proliferation in HEC-1A and Ishikawa cells. However, SESN2 mRNA and protein expression was upregulated in endometrial cancer patient tissues where mTORC1 activity was still high. The results from the GEPIA database indicated statistically significant positive correlations of expression of SESN2 with CDKN1A and CDKN1B. Our data thus suggest that these correlations between SESN2 and cell cycle-associated genes are independent of the role of CDKN1A and CDKN1B as tumor suppressors in the tumor of patients with endometrial cancer.

SESN2 suppresses migration and proliferation in vitro, and tumor growth in vivo in colorectal cancer [27,55,56,57]. However, the relationship between growth and SESN2 expression has remained unanswered in endometrial cancer. SESN2 expression is higher in endometrial cancer tissues than normal endometrial tissues, and its knockdown stimulated endometrial cancer cell proliferation and tumor growth in vivo, as observed in a mouse xenograft model. Additionally, *SESN2* knockdown promoted EMT and migration of endometrial cancer cells. We also found that overexpression of *SESN2* inhibited both expression of N-cadherin and activation of p70S6K. The anti-cancer effects of SESN2 are associated with regulation of the mTORC1 signaling pathway [58,59]. The mTOR signaling pathway is hyperactivated during carcinogenesis and tumor progression [60]. Activation of mTOR increases protein synthesis, and chronic mTOR activation causes ER stress [12]. To assess whether the anti-cancer effect of SESN2 on endometrial cancer cells is regulated via the mTORC1 pathway, we treated *SESN2* knocked-down HEC-1A and Ishikawa cells with rapamycin and Torin 1. Treatment with both agents reversed endometrial cancer cell proliferation, migration, and EMT marker expression in the *SESN2* knocked-down cells. Moreover, we found that mTORC1 activation by *TSC2* knockdown led to a reduction in E-cadherin expression and an increase in N-cadherin expression. Thus, the analysis suggests that the inhibitory effect of SESN2 on proliferation and migration of endometrial cancer cells is dependent on mTORC1. Additionally, SESN2 suppresses tumor growth by enhancing autophagy activity through the inhibition of the mTORC1 pathway [30,61]. We have probed the mTORC1-dependent mechanism of SESN2 that suppresses endometrial cancer cells, although further experimental studies would be required to explore the role of SESN2-mediated autophagy in inhibiting endometrial cancer growth and progression.

## 4. Materials and Methods

### 4.1. Reagents

Antibodies used for immunoblotting included anti-sestrin2 obtained from Proteintech Group, anti-phospho-p70 S6 Kinase, anti-phospho-S6 Ribosomal Protein, anti-S6 Ribosomal Protein, anti-E-cadherin, anti-N-cadherin, and anti-TSC2 from Cell Signaling Technology, anti-p70 S6 Kinase from Santa Cruz Biotechnology, anti-glyceraldehyde-3-phosphate dehydrogenase (GAPDH) from Aviva Systems Biology, and anti-β-Actin from Developmental Studies Hybridoma Bank. Rapamycin was obtained from LC Laboratories. Torin 1 was purchased from Cayman Chemical. N-acetyl cysteine (NAC) was purchased from Sigma-Aldrich.

### 4.2. Patient Tissue Samples

Tissue samples of endometrial cancer and normal endometrium were collected from 6 patients with endometrial cancer and 5 patients with myoma of the uterus respectively, who underwent surgical resection at the Soonchunhyang University Seoul Hospital. Fresh specimens were snap-frozen and stored at –80 °C. Informed consent was obtained from all the patients before sample collection. The study was approved by the Institutional Review Board of the Soonchunhyang University Seoul Hospital (IRB no. 2016-06-008).

### 4.3. GEPIA Database Analysis

The Gene Expression Profiling Interactive Analysis (GEPIA; http://gepia.cancer-pku.cn/index.html) [62] is a web server to perform customizable analyses of RNA sequencing expression data of 8587 normal and 9736 tumor samples from the Cancer Genome Atlas (TCGA) and the Genotype-Tissue Expression (GTEx) project. GEPIA was used to generate survival curves, including overall survival and disease-free survival, based on SESN2 expression, and log-rank test in endometrial cancer. In addition, we analyzed the correlation between expression of SESN2 and other genes related to our study through related modules in GEPIA.

### 4.4. UCSC Cancer Genomics Browser Analysis

The data of SESN2 expression in patients with endometrial cancer were obtained from the TCGA-Uterine Corpus Endometrial Carcinoma (UCEC). Original data were downloaded from the University of California at Santa Cruz (UCSC) Cancer Genomics Browser (http://xena.ucsc.edu/) [63]. Further, we used the UCSC Cancer Genomics Browser to analyze the expression of the SESN2 gene.

### 4.5. Cell Culture and Treatments

Human endometrial adenocarcinoma cell lines HEC-1A and Ishikawa were obtained from the ATCC (Rockville, MD, USA) and Dr. Sung Ki Lee (Konyang University Hospital), and cultured in Dulbecco’s Modified Eagle’s Medium (DMEM, Welgene, Daegu, Korea) containing 10% fetal bovine serum (FBS, Welgene), 4.5 g/L D-glucose with L-glutamine, and 100 U/mL penicillin-streptomycin (Welgene). The human embryonic kidney (HEK) 293T cells, a packaging cell line for lentivirus production, were cultured in DMEM containing 10% FBS, 4.5 g/L D-glucose with L-glutamine, 110 mg/L sodium pyruvate, and 100 U/mL penicillin-streptomycin. Both the cultures were maintained in a humidified 5% CO_2_ atmosphere at 37 °C. For mTOR inhibitor treatment, the HEC-1A and Ishikawa cells were incubated in the presence of 100 nM rapamycin or 250 nM Torin 1. The same volume of dimethyl sulfoxide was used as vehicle control.

### 4.6. Viral Transduction

HEK-293T cells were transfected with the following lentiviral packaging constructs using polyethylenimine reagent: sh-luciferase (sh-*LUC*), sh-*SESN2*, sh-*TSC2*, cytomegalovirus promoter-green fluorescence protein (CMV-GFP), and CMV-Flag-h*SESN2* constructs (a kind gift by Dr. A. V. Budanov, Trinity College, Ireland) [12]. Lentiviral supernatants were collected and filtered 48 and 72 h after transfection. The HEC-1A and Ishikawa cells were incubated for 48 h with lentiviral medium in the presence of 4 µg/mL polybrene.

### 4.7. Quantitative Real-Time PCR

Total RNA was extracted from endometrial tissues, HEC-1A, and Ishikawa cells using NucleoZOL reagent (Takara) according to the manufacturer’s instructions. Complementary DNA was synthesized using Moloney murine leukemia virus reverse transcriptase (MMLV-RT, Promega) and random hexamers (BioFact). Quantitative real-time reverse transcription PCR (qRT-PCR) analysis was performed in triplicates using the SYBR green real-time PCR master mix reagent (Biofact) and QuantStudio 3 Real-time PCR System (Life Technologies Inc.). Relative mRNA expression was calculated from the comparative threshold cycle (C_t_) values relative to human *GAPDH*. The following primers were used—*GAPDH:* forward 5′-TTGCCATCAATGACCCCTTCA-3′, reverse 5′-CGCCCCACTTGATTTTGGA-3′; *Cyclophilin:* forward 5′-GCAAAGTGAAAGAAGGCA-3′, reverse 5′-CCATTCCTGGACCCAAAG-3′; *SESN2:* forward 5′-ACTGCGTCTTTGGCATCAG-3′, reverse 5′-CTTCTCTGAGTGGCGGAAGT-3′; *MKI67:* forward 5′-ACGCCTGGTTACTATCAAAAG-3′, reverse 5′-CAGACCCATTTACTTGTGTTGGA-3′; *CDKN1A:* forward 5′-TGTCCGTCAGAACCCATGC-3′, reverse 5′-AAAGTCGAAGTTCCATCGCTC-3′; and *CDKN1B:* forward 5′-AACGTGCGAGTGTCTAACGG-3′, reverse 5′-CCCTCTAGGGGTTTGTGATTCT-3′.

### 4.8. Immunoblotting

Endometrial cancer tissues, HEC-1A, and Ishikawa cells were lysed in an ice-cold radio-immunoprecipitation assay buffer containing complete protease inhibitor cocktail (Roche). The lysates were incubated for 20 min on ice and centrifuged at 18,000× *g* for 15 min at 4 °C. Protein concentration was measured using the bicinchoninic acid (BCA) protein assay (Pierce). The lysates were boiled in sodium dodecyl sulfate (SDS) Laemmli sample buffer for 5 min, resolved using SDS-polyacrylamide gel electrophoresis, transferred onto polyvinylidene fluoride membranes (Millipore), and probed with primary antibodies against SESN2, p-p70S6K, p70S6K, p-S6, S6, E-cadherin, N-cadherin, TSC2, GAPDH, α-tubulin, or β-actin. After sample incubation with secondary antibodies conjugated with horseradish peroxidase (Bio-Rad, Hercules, CA, USA), chemiluminescence signals were detected using the Fusion Solo System (Vilber Lourmat, Marne-la-VallCe, France). Densitometric analysis of the blots was performed using ImageJ software (National Institutes of Health), with which the background was removed for each band.

### 4.9. Cell Proliferation Assay

HEC-1A and Ishikawa cells were seeded in 12-well plates at a density of 1 × 10^4^ cells/well and incubated for 4 days in normal growth medium. The HEC-1A and Ishikawa cells were harvested using trypsin-ethylenediaminetetraacetic acid (EDTA) solution and washed twice with phosphate-buffered saline (PBS). Cell numbers were measured by counting viable cells using hemocytometer under an inverted microscope (Motic AE 2000) after trypan blue staining.

### 4.10. Detection of Reactive Oxygen Species (ROS)

The intracellular levels of ROS were analyzed using the ROS-reactive fluorescent indicator 5-(and-6)-chloromethyl-2′,7′-dichlorodihydrofluorescein diacetate, acetyl ester (CM-H_2_DCFDA, Invitrogen), according to the manufacturer’s instructions. The HEC-1A and Ishikawa cells were seeded on a μ-Slide four-well chamber slide (Ibidi), treated with 5 µM of CM-H_2_DCFDA for 30 min, and washed with PBS. Samples were analyzed under a fluorescence microscope (Eclipse TS2, Nikon, Tokyo, Japan).

### 4.11. Wound-Healing Assay

Cell migration was assessed by wound-healing assay. HEC-1A and Ishikawa cells transduced with lentiviral vectors expressing luciferase control shRNA or *SESN2*-shRNA were seeded into 6-well plates and cultured to 90% confluence. The confluent cell monolayer was wounded by manual scratching with a 200 μL pipette tip. The cells were washed with PBS and incubated at 37 °C in normal growth medium. Phase contrast images at specific wound sites were obtained at the indicated time points using an inverted microscope.

### 4.12. In Vivo Tumorigenesis

All animal care and handling protocols were approved by the Institutional Animal Care and Use Committee of Konyang University (approval no. 20-25-A-02). Five-week-old female BALB/C athymic nude mice were maintained and treated under pathogen-free conditions. HEC-1A cells transduced with lentiviral vectors expressing luciferase control shRNA or *SESN2*-shRNA were cultured for at least 10 days in puromycin (0.5 µg/mL). After selection, the stable expressing cells (5 × 10^6^ cells) of sh-*LUC* or sh-*SESN2* suspended in 100 μL PBS were injected into the bilateral hind leg subcutaneous tissue of the mice. The tumor volumes were measured every week. Tumor volume (mm^3^) was calculated by the following formula: length × width × height × 0.52. Six weeks later, the nude mice were euthanized, and their tumors were excised and weighed.

### 4.13. Statistical Analysis

Results are presented as the mean ± standard error of mean (SEM). The data presented in the figure panels are representative of at least three independent experiments, unless otherwise mentioned. The significance of differences between two experimental groups has been determined using a two-tailed Student’s *t*-test. A *p*-value < 0.05 was considered statistically significant (* *p* < 0.05; ** *p* < 0.01; *** *p* < 0.001).

## 5. Conclusions

In summary, this study provided evidence for the prognostic significance and potential therapeutic role of SESN2 in endometrial cancer. The analysis suggested increased expression of SESN2 in endometrial cancer tissues, which correlates with high mTORC1 activity. SESN2 was found to regulate endometrial cancer growth, migration, and ROS production via the mTORC1-dependent mechanism. This suggested SESN2 to function as a negative feedback regulator of mTORC1 to inhibit endometrial cancer growth and progression stimulated by hyperactivation of mTOR signaling. Thus, our study elucidates the relationship between SESN2, the mTORC1 pathway, and endometrial cancer growth, and defines SESN2 as a therapeutic target to inhibit the progression of endometrial cancer.

## Figures and Tables

**Figure 1 cancers-12-02515-f001:**
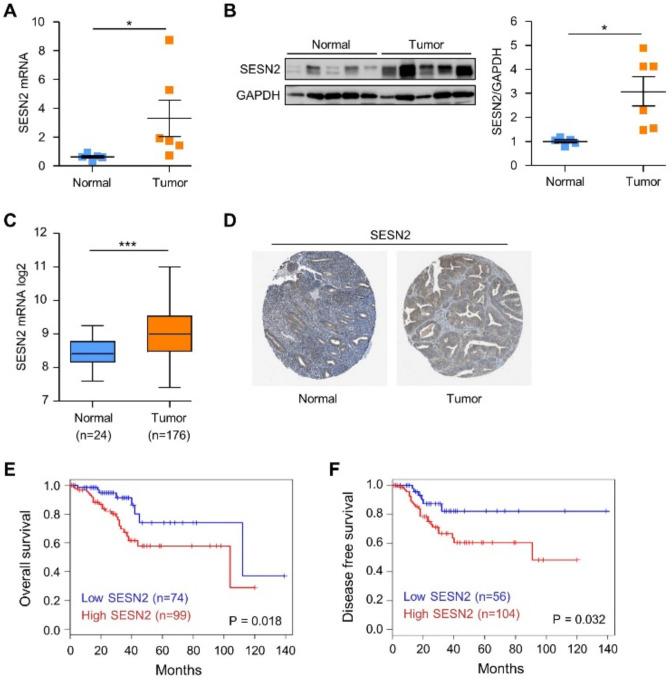
The expression and clinical significance of Sestrin2 (SESN2) in endometrial cancer. (**A**) Relative mRNA expression levels of *SESN2* in endometrial cancer (*n* = 6) and normal endometrium (*n* = 5). The relative mRNA levels of *SESN2* in each sample are normalized to that of *GAPDH*. (**B**) Immunoblot analysis of SESN2 in endometrial cancer (*n* = 6) and normal endometrium (*n* = 5). GAPDH served as an internal loading control; band intensities are quantified and normalized to GAPDH values. (**C**) *SESN2* gene expression in endometrial cancer (*n* = 176) and normal endometrium (*n* = 24) samples. TCGA data was downloaded from UCSC Xena portal. Data are shown as mean ± SEM. * *p* < 0.05; *** *p* < 0.001 (Student’s *t*-test). (**D**) SESN2 protein expression in endometrial cancer and normal endometrial tissue specimens. Images were obtained from the Human Protein Atlas online database. (**E**,**F**) Kaplan–Meier survival curves based on SESN2 expression levels were analyzed from the GEPIA database. Overall survival (**E**) and disease-free survival (**F**) curves of endometrial cancer patients are indicated with high SESN2 (red line) versus low SESN2 expression (blue line). The *p*-values for estimating significance of differences between high and low expression were calculated using the log-rank test.

**Figure 2 cancers-12-02515-f002:**
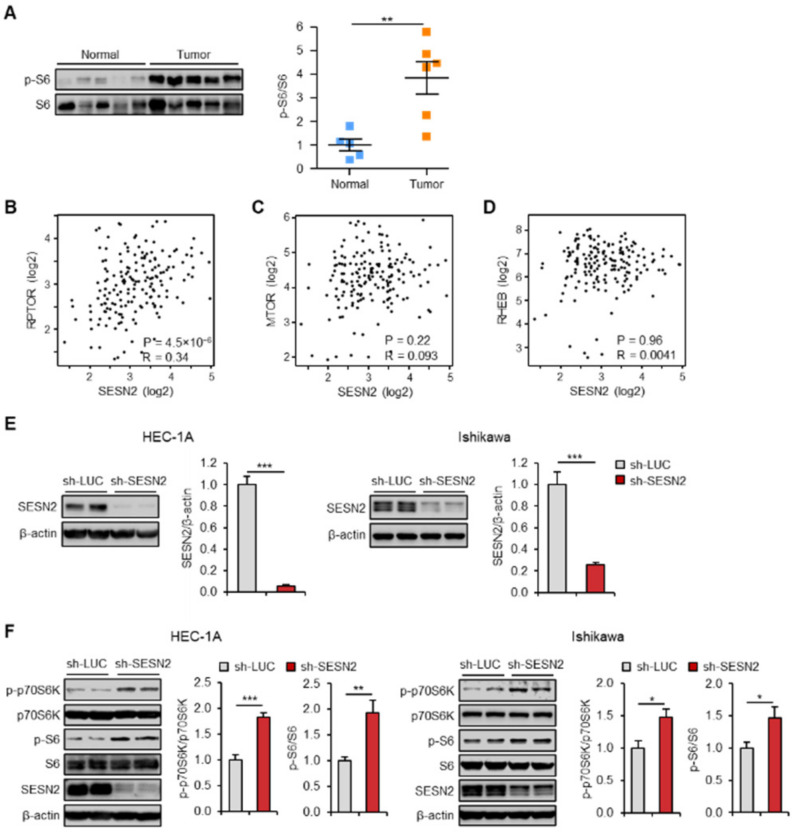
Correlation of expression of SESN2 with the mTOR pathway in endometrial cancer. (**A**) Immunoblot analysis of p-S6 in endometrial cancer (*n* = 6) and normal endometrium (*n* = 5). S6 served as a loading control. Band intensities are quantified and normalized with S6 values. (**B**–**D**) Correlation between *SESN2* RNA expression and with those of mTOR pathway-related markers, including regulatory associated protein of mTOR (*RPTOR*) (**B**), *MTOR* (**C**), and *RHEB* (**D**), in endometrial cancer analyzed using the GEPIA database. The correlation coefficients were calculated using Spearman’s test. (**E**) Immunoblotting analysis of SESN2 in HEC-1A and Ishikawa cells infected with lentiviruses expressing shRNAs targeting luciferase (sh-*LUC*) or *SESN2* (sh-*SESN2*). Cell lysates were immunoblotted with anti-SESN2 antibody. β-actin served as a loading control. Band intensities are quantified and normalized over the β-actin values. (**F**) Immunoblotting analysis of phospho-p70 ribosomal protein S6 kinase (p-p70S6K), p70S6K, p-S6, and p6 in HEC-1A and Ishikawa cells were infected with sh-*LUC* or sh-*SESN2*. Cell lysates were immunoblotted with anti-p-p70S6K, anti-p70S6K, anti-p-S6, and anti-S6 antibodies. β-actin served as a loading control. Band intensities were quantified and normalized with p70S6K or S6 values. Data are shown as mean ± SEM. Results are representative of at least three independent experiments. * *p* < 0.05; ** *p* < 0.01; *** *p* < 0.001 (Student’s *t*-test).

**Figure 3 cancers-12-02515-f003:**
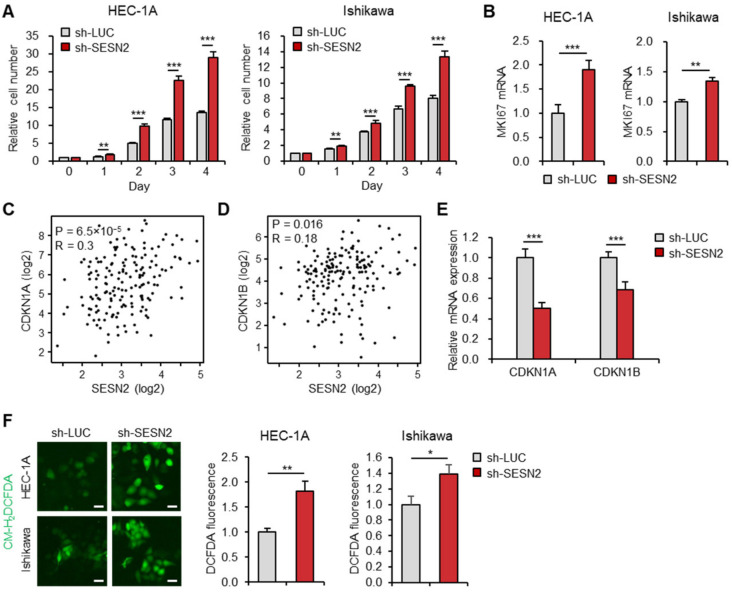
Knockdown of *SESN2* promotes proliferation and ROS production in endometrial cancer cells. (**A**) Cell proliferation analysis in HEC-1A and Ishikawa cells infected with lentiviruses expressing shRNAs targeting luciferase (sh-*LUC*) or *SESN2* (sh-*SESN2*). (**B**) qRT-PCR analysis of *MKI67* mRNA levels in HEC-1A and Ishikawa cells infected with sh-*LUC* or sh-*SESN2* lentivirus. (**C**,**D**) Correlation between expression of *SESN2* and cell cycle inhibitor genes *CDKN1A* (**C**) and *CDKN1B* (**D**) in endometrial cancer analyzed using the GEPIA database. The correlation coefficients were calculated using Spearman’s test. (**E**) Relative mRNA expression of *CDKN1A* and *CDKN1B* in HEC-1A cells infected with sh-*LUC* or sh-*SESN2* lentivirus. (**F**) Intracellular ROS levels were analyzed by the CM-H_2_DCFDA probe in HEC-1A and Ishikawa cells infected with sh-*LUC* or sh-*SESN2* lentivirus. Data are shown as mean ± SEM. Results are representative of at least three independent experiments. * *p* < 0.05; ** *p* < 0.01; *** *p* < 0.001 (Student’s *t*-test).

**Figure 4 cancers-12-02515-f004:**
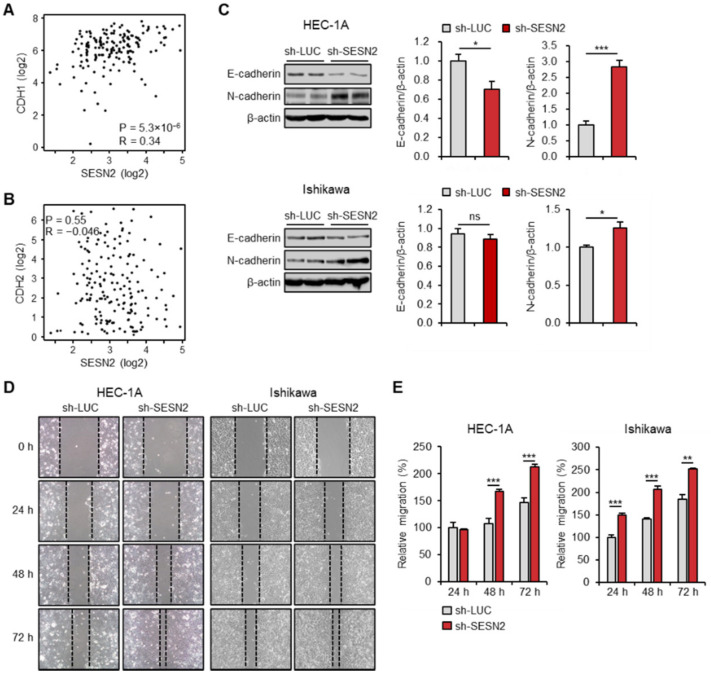
Knockdown of *SESN2* promotes EMT and migration in endometrial cancer cells. (**A**,**B**) Correlation between expression of *SESN2* and EMT marker genes *CDH1* (coded E-cadherin) (**A**) and *CDH2* (coded N-cadherin) (**B**) in endometrial cancer, analyzed using the GEPIA database. The correlation coefficients were calculated using Spearman’s test. (**C**) Immunoblotting analysis in HEC-1A and Ishikawa cells infected with sh-*LUC* or sh-*SESN2* lentivirus. Cell lysates were immunoblotted with anti-E-cadherin and anti-N-cadherin antibodies. β-actin served as a loading control. Band intensities were quantified and normalized with β-actin values. (**D**) Scratch assays for HEC-1A and Ishikawa cells infected with lentiviruses expressing shRNAs targeting luciferase (sh-*LUC*) or *SESN2* (sh-*SESN2*). Monolayers of cells were scratch-wounded with a sterile tip and photographed using a phase-contrast microscope at 0, 24, 48, and 72 h after wounding. (**E**) Quantification of wound migration in (**D**). Data are shown as mean ± SEM. Results are representative of at least three independent experiments. * *p* < 0.05; ** *p* < 0.01; *** *p* < 0.001 (Student’s *t*-test).

**Figure 5 cancers-12-02515-f005:**
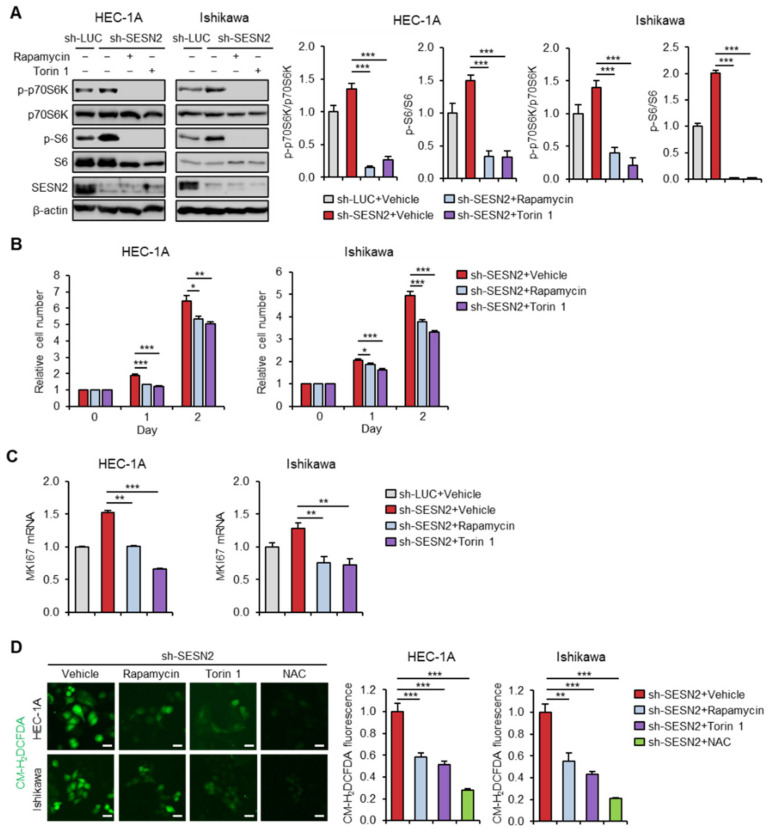
Knockdown of *SESN2* increases endometrial cancer cell proliferation and ROS production in a mTOR-dependent manner. (**A**) Immunoblotting analysis for mTORC1-related proteins in HEC-1A and Ishikawa cells infected with lentiviruses expressing shRNAs targeting luciferase (sh-*LUC*) or *SESN2* (sh-*SESN2*) and treated with 100 nM rapamycin or 250 nM Torin 1 for 24 h. Cell lysates were immunoblotted with anti-p-p70S6K, anti-p70S6K, anti-p-S6, anti-S6, and anti-SESN2 antibodies. β-actin served as a loading control. Band intensities were quantified and normalized with p70S6K or S6 values. (**B**) Cell proliferation analysis in HEC-1A and Ishikawa cells infected with sh-*LUC* or sh-*SESN2* lentivirus and treated with 100 nM rapamycin or 250 nM Torin 1. (**C**) qRT-PCR analysis of *MKI67* mRNA levels in HEC-1A and Ishikawa cells infected with sh-*LUC* or sh-*SESN2* lentivirus and treated with 100 nM rapamycin or 250 nM Torin 1 for 6 h. (**D**) Intracellular ROS levels were analyzed by CM-H_2_DCFDA in HEC-1A and Ishikawa cells infected with sh-*LUC* or sh-*SESN2* lentivirus and treated with 100 nM rapamycin, 250 nM Torin 1, or 5 mM N-acetyl cysteine (NAC) for 6 h. Data are shown as mean ± SEM. Results are representative of at least three independent experiments. * *p* < 0.05; ** *p* < 0.01; *** *p* < 0.001 (Student’s *t*-test).

**Figure 6 cancers-12-02515-f006:**
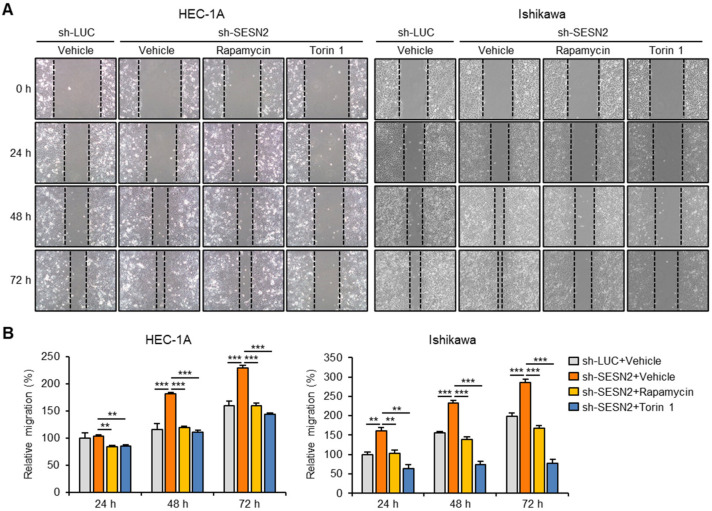
Knockdown of *SESN2* increases migration of endometrial cancer cells in a mTOR-dependent manner. (**A**) Wound scratch assay in HEC-1A and Ishikawa cells infected with lentiviruses expressing shRNAs targeting luciferase (sh-*LUC*) or *SESN2* (sh-*SESN2*) and treated with 100 nM rapamycin or 250 nM Torin 1. Monolayers of cells were scratch-wounded with a sterile tip and photographed using a phase-contrast microscope at 0, 24, 48, and 72 h after wounding. (**B**) Quantification of wound migration in (**A**). Data are shown as mean ± SEM. Results are representative of at least three independent experiments. ** *p* < 0.01; *** *p* < 0.001 (Student’s *t*-test).

**Figure 7 cancers-12-02515-f007:**
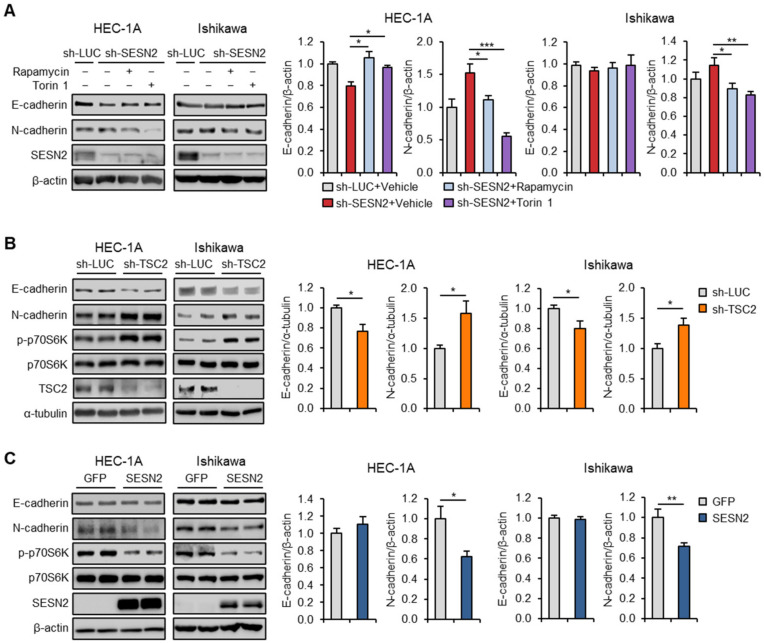
SESN2 regulates E-cadherin and N-cadherin via the mTORC1 pathway in HEC-1A and Ishikawa cells. (**A**) Immunoblotting analysis of E-cadherin, N-cadherin, and SESN2 in HEC-1A and Ishikawa cells infected with lentiviruses expressing shRNAs targeting luciferase (sh-*LUC*) or *SESN2* (sh-*SESN2*) and treated with 100 nM rapamycin or 250 nM Torin 1 for 24 h. Cell lysates were immunoblotted with anti-E-cadherin, anti-N-cadherin, and anti-SESN2 antibodies. (**B**) Immunoblotting analysis of E-cadherin, N-cadherin, and mTORC1-related proteins in HEC-1A and Ishikawa cells infected with sh-*LUC* or sh-*TSC2* lentivirus. Cell lysates were immunoblotted with anti-E-cadherin, anti-N-cadherin, anti-p-p70S6K, anti-p70S6K, and anti-TSC2 antibodies. (**C**) Immunoblotting analysis in HEC-1A and Ishikawa cells infected with lentiviruses overexpressing green fluorescent protein (GFP) as control or *SESN2*. Cell lysates were immunoblotted with anti-E-cadherin, anti-N-cadherin, anti-p-p70S6K, anti-p70S6K, and anti-SESN2 antibodies. β-Actin or α-tubulin served as a loading control. Band intensities were quantified and normalized with β-actin or α-tubulin values. Data are shown as mean ± SEM. Results are representative of at least three independent experiments. * *p* < 0.05; ** *p* < 0.01; *** *p* < 0.001 (Student’s *t*-test).

**Figure 8 cancers-12-02515-f008:**
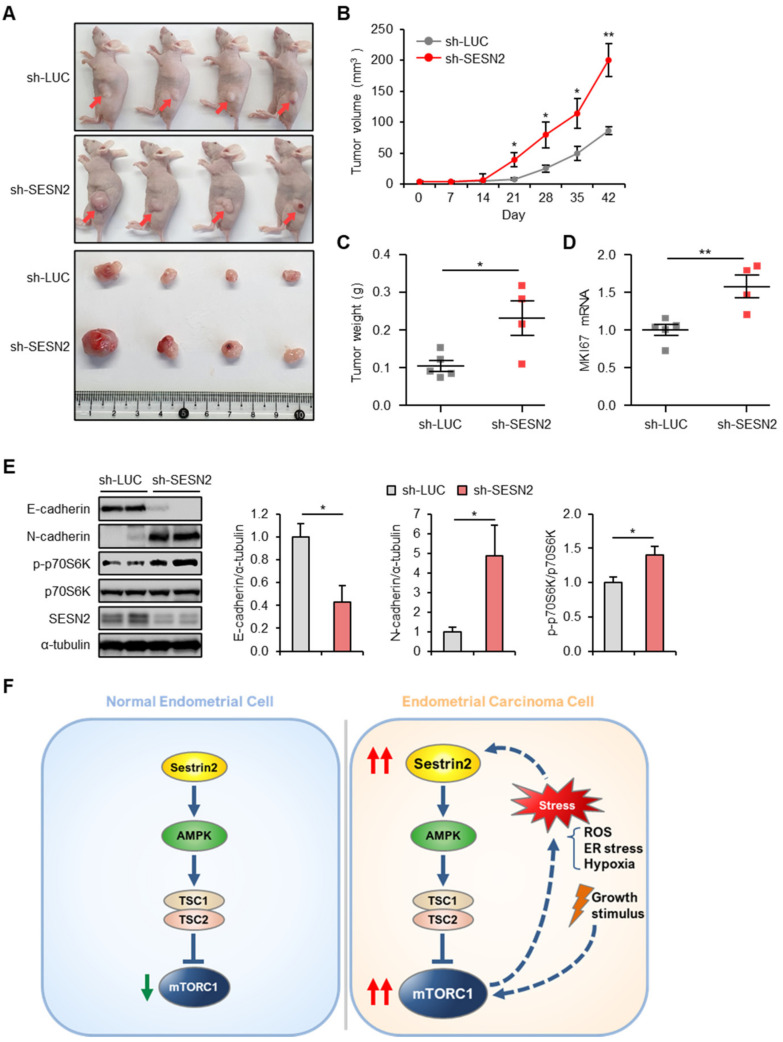
In vivo effects of SESN2 on tumor growth. (**A**–**E**) HEC-1A cells infected with lentiviruses expressing shRNAs targeting luciferase (sh-LUC, *n* = 5) or SESN2 (sh-SESN2, *n* = 4) were subcutaneously injected into nude mice. (**A**) A representative photograph of nude mice at 6 weeks after tumor cell injection. Red arrows indicate the xenograft tumor. (**B**) Tumor growth curve is plotted based on the tumor volume. (**C**) Differences in tumor weight measured while excising the tumors. (**D**) qRT-PCR analysis of *MKI67* mRNA levels in tumor tissues from the indicated mice. (**E**) Immunoblotting analysis in tumor lysates from indicated mice with anti-E-cadherin, anti-N-cadherin, anti-p-p70S6K, anti-p70S6K, and anti-SESN2 antibodies. The α-Tubulin served as a loading control. Band intensities were quantified and normalized with α-tubulin values. (**F**) Schematic diagram describing the roles and mechanisms of SESN2 in endometrial cancer. Dashed arrows indicate speculations based on current and previous studies. Data are shown as mean ± SEM. * *p* < 0.05; ** *p* < 0.01 (Student’s *t*-test).

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
