# Peer review of "mTOR-Dependent Role of Sestrin2 in Regulating Tumor Progression of Human Endometrial Cancer"

_cancers, 2020, doi:10.3390/cancers12092515_

Round 1
Reviewer 1 Report
In this study, Shin et al. explored the role of SESN2 in the progression of endometrial cancer and its correlation with mTOR signaling. Overall, the manuscript is very well prepared and the experiments were properly performed. The in vitro results are very well presented and the discussed. However, the authors also presented results of samples from patients with endometrial cancer and the observations are in disagreement with the cell lines results. While in cell lines SESN2 seems to work as a tumor suppressor, regulating mTOR signaling, ROS levels, proliferation, migration and EMT; the results from patients indicate that SESN2 accumulation is associated with reduced survival and increased mTOR levels. These conflicting results were not properly addressed in the discussion. Also, figure 8F is not properly summarizing the results, since the authors did not presented any results indicating the regulation of SESN2 by stress, and high levels of SESN2 diminished mTOR only in cell lines, but the same is not occurring in patients.
In my opinion, the authors should re-think the discussion to better address the results.
Minor comments:
1- Introduction - lines 79-80: "In this study, for the first time, we examined SESN2 expression levels in the primary tumor of a patient with endometrial cancer and its correlation with clinicopathological factors."
The authors evaluated samples from 6 patients and 176 from TCGA database, not one. Please, correct the sentence.
2- Figure 7A: immunoblotting results from e-cadherin of Ishikawa cell line seem different from figure 4C. Also, the graphic representing the quantification shows a different result than the observed in the immunoblotting. I would suggest the authors to replace the image.
Author Response
Dear Reviewers,
We highly appreciate the reviewers for constructive criticisms and valuable comments. Each comment has been carefully considered point by point and responded. Specific responses to each of the comments are provided below.
Reviewer #1:
In this study, Shin et al. explored the role of SESN2 in the progression of endometrial cancer and its correlation with mTOR signaling. Overall, the manuscript is very well prepared and the experiments were properly performed. The in vitro results are very well presented and the discussed. However, the authors also presented results of samples from patients with endometrial cancer and the observations are in disagreement with the cell lines results. While in cell lines SESN2 seems to work as a tumor suppressor, regulating mTOR signaling, ROS levels, proliferation, migration and EMT; the results from patients indicate that SESN2 accumulation is associated with reduced survival and increased mTOR levels. These conflicting results were not properly addressed in the discussion. Also, figure 8F is not properly summarizing the results, since the authors did not presented any results indicating the regulation of SESN2 by stress, and high levels of SESN2 diminished mTOR only in cell lines, but the same is not occurring in patients.
In my opinion, the authors should re-think the discussion to better address the results.
Response: We thank the reviewer for the constructive suggestion. We agree that our discussion need to address our results better. High SESN2 expression was associated with poor prognosis in endometrial cancer patients, whereas knockdown of SESN2 promoted cell proliferation and migration in endometrial cancer cell lines HEC-1A and Ishikawa. Although SESN2 expression was correlated with mTOR signaling pathway, SESN2 may not be the direct cause of increased mTORC1 activity, but rather the opposite—that increased mTORC1 activity caused the increase in SESN2. In endometrial cancer patient tissues, inhibition of mTORC1 activity was not effectively mediated by the negative feedback regulation of SESN2, whereas in endometrial cancer cell lines, lentiviral overexpression of SESN2 effectively suppressed mTORC1 activity. A possible reason for the disagreement between patient and cell line data may be due to the bidirectional role of SESN2 in endometrial cancer progression. Activation of multiple stresses, including ER stress and oxidative stress, by dysregulated activation of mTORC1, initiates a negative-feedback loop to suppress mTORC1 activity by transcriptional activation of SESN2 during cancer cell growth and expansion (Park, Park et al. 2014, Heberle, Prentzell et al. 2015, D'Orazi and Cirone 2019). We found that SESN2 knockdown enhanced mTORC1 activation in endometrial cancer cell lines. Moreover, overexpression of SESN2 reduced mTORC1 activation in endometrial cancer cell lines. However, mTORC1 activity was significantly higher in endometrial cancer tissues of the patients than in normal endometrial tissues despite a significant increase in SESN2 expression. These findings may be the result of multiple molecular events that can lead to hyperactivation of mTORC1 during endometrial cell growth and expansion (Yecies and Manning 2011, Heberle, Prentzell et al. 2015). Thus, it appears that poor prognosis in endometrial cancer patients is caused by hyperactivation of the mTORC1 pathway rather than by high SESN2 expression. As the reviewer suggested, we have now included this statement in the revised Discussion section [marked] (page 12 line 348-367).
As mentioned earlier, cancer cells undergo various forms of intrinsic stress and adverse environmental challenges, such as oxidative stress, ER stress, proteotoxic, inflammatory stress, and nutrient deprivation, that try to manage by activating molecular/cellular pathways (D'Orazi and Cirone 2019). Although SESN2 is critically involved in cellular responses to various stresses and has a protective effect on physiological and pathological states, we agree that it is not certain that mechanistic linkage by which Activation of multiple stresses by dysregulated activation of mTORC1, initiates a negative-feedback loop by transcriptional activation of SESN2. We have therefore revised the new Figure 8F and the Figure legend as suggested in order to make sure we are not over-interpreting our findings or drawing conclusions that are not substantiated by the evidence.
Minor comments:
- Introduction - lines 79-80: "In this study, for the first time, we examined SESN2 expression levels in the primary tumor of a patient with endometrial cancer and its correlation with clinicopathological factors."
The authors evaluated samples from 6 patients and 176 from TCGA database, not one. Please, correct the sentence.
Response: We are very thankful for the reviewer’s comments. Accordingly, the sentence was corrected in the revised manuscript as following (page 2 line 78-80): we examined SESN2 expression levels in the primary tumors of endometrial cancer patients and in the endometrial cancer samples of The Cancer Genome Atlas (TCGA), and their correlations with clinicopathological factors.
- Figure 7A: immunoblotting results from e-cadherin of Ishikawa cell line seem different from figure 4C. Also, the graphic representing the quantification shows a different result than the observed in the immunoblotting. I would suggest the authors to replace the image.
Response: As requested by the reviewer, we have replaced it with a new western blot image (new Figure 7A).
Reviewer 2 Report
In the present manuscripts, the authors reported that Sestrin2 (SESN2) was upregulated in endometrial cancer cells and tissues and correlated with increased mTORC1 activity. In addition, upregulation of SESN2 was associated with reduced overall and disease-free survival in patients in endometrial cancer. However, knockdown of SESN2 increased cell proliferation, migration and ROS production, and mTOR inhibitors reversed the effects of SESN2 knockdown. The authors concluded that SESN2 is a potential therapeutic target in endometrial cancer. While the results support that knockdown of SESN2 enhances cell proliferation, migration and tumor growth by activating mTORC1, there are several contradictions that need to be resolved.
Comments:
- Although increased SESN2 expression is associated with poor patient survival, knockdown of SESN2 increased cell proliferation, migration and tumor growth. Thus, it is not clear how SESN2 could serve as a potential therapeutic target in endometrial cancer.
- Figures 2B, C and D add little to the manuscript since the functional significance of the correlation between SESN2 and RPTOR and lack of correlation between SESN2 and mTOR or RHEB was not addressed.
- 3B: How does SESN2 regulate MKI67 mRNA expression?
- The authors mentioned that SESN2 expression positively correlates with mTORC1 activity (line 31), which increases cell proliferation and is increased in endometrial cancers (Fig. 1A-1D). However, SESN2 expression was also positively correlated with cyclin-dependent kinase inhibitors CDKN1A and CDKN1B (line 171-172, Fig. 3C & D), which inhibit cell proliferation. The authors should provide an explanation.
- Figure 4C: SESN2 knockdown had little effect on E-cadherin in Ishikawa cells.
- Figure 8F: While previous studies have shown that transcriptional activation of SESN2 is one of the negative feedback mechanisms for inhibiting chronic activation of mTORC1 (line 69-71), it is not clear if SESN2 also negatively regulates mTORC1 via AMPK in endometrial cancers.
- Line 74 and 75: It was mentioned that SESN2 expression is cell-type specific. But it seems that SESN2 expression was reported to be both downregulated and upregulated in hepatocellular carcinoma.
- 321: Reference should be cited for limited studies discussing the regulation of the mTOR pathway.
Author Response
Dear Reviewers,
We highly appreciate the reviewers for constructive criticisms and valuable comments. Each comment has been carefully considered point by point and responded. Specific responses to each of the comments are provided below.
In the present manuscripts, the authors reported that Sestrin2 (SESN2) was upregulated in endometrial cancer cells and tissues and correlated with increased mTORC1 activity. In addition, upregulation of SESN2 was associated with reduced overall and disease-free survival in patients in endometrial cancer. However, knockdown of SESN2 increased cell proliferation, migration and ROS production, and mTOR inhibitors reversed the effects of SESN2 knockdown. The authors concluded that SESN2 is a potential therapeutic target in endometrial cancer. While the results support that knockdown of SESN2 enhances cell proliferation, migration and tumor growth by activating mTORC1, there are several contradictions that need to be resolved.
Comments:
- Although increased SESN2 expression is associated with poor patient survival, knockdown of SESN2 increased cell proliferation, migration and tumor growth. Thus, it is not clear how SESN2 could serve as a potential therapeutic target in endometrial cancer.
Response: We thank the reviewer for the constructive suggestion. We agree that our discussion need to address our results better. High SESN2 expression was associated with poor prognosis in endometrial cancer patients, whereas knockdown of SESN2 promoted cell proliferation and migration in endometrial cancer cell lines HEC-1A and Ishikawa. Although SESN2 expression was correlated with mTOR signaling pathway, SESN2 may not be the direct cause of increased mTORC1 activity, but rather the opposite—that increased mTORC1 activity caused the increase in SESN2. In endometrial cancer patient tissues, inhibition of mTORC1 activity was not effectively mediated by the negative feedback regulation of SESN2, whereas in endometrial cancer cell lines, lentiviral overexpression of SESN2 effectively suppressed mTORC1 activity. A possible reason for the disagreement between patient and cell line data may be due to the bidirectional role of SESN2 in endometrial cancer progression. Activation of multiple stresses, including ER stress and oxidative stress, by dysregulated activation of mTORC1, initiates a negative-feedback loop to suppress mTORC1 activity by transcriptional activation of SESN2 during cancer cell growth and expansion (Park, Park et al. 2014, Heberle, Prentzell et al. 2015). We found that SESN2 knockdown enhanced mTORC1 activation in endometrial cancer cell lines. Moreover, overexpression of SESN2 reduced mTORC1 activation in endometrial cancer cell lines. However, mTORC1 activity was significantly higher in endometrial cancer tissues of the patients than in normal endometrial tissues despite a significant increase in SESN2 expression. These findings may be the result of multiple molecular events that can lead to hyperactivation of mTORC1 during endometrial cell growth and expansion (Yecies and Manning 2011, Heberle, Prentzell et al. 2015). Thus, it appears that poor prognosis in endometrial cancer patients is caused by hyperactivation of the mTORC1 pathway rather than by high SESN2 expression.
SESN2 deficiency in endometrial cancer tissues may lead to higher mTORC1 activity, which is associated with poor prognosis. On the other hand, exogenous SESN2 gene expression at therapeutic levels may suppress endometrial cancer progression by inhibiting mTORC1 activity. Accordingly, we have now included this statement in the revised Discussion section [marked] (page 12 line 348-367).
- Figures 2B, C and D add little to the manuscript since the functional significance of the correlation between SESN2 and RPTOR and lack of correlation between SESN2 and mTOR or RHEB was not addressed.
Response: We appreciate the reviewer’s comment. mTORC1 is composed of mTOR, raptor (regulatory associated protein of TOR), mLST8, PRAS40, and deptor (DEP domain containing mTOR interacting protein). Rheb GTPase directly binds to mTOR and activates mTORC1 in response to growth factor signals (Kim and Guan 2019). Reasons for the correlations between SESN2 and MTOR or RHEB being not significant could be because mRNA levels of the core components of mTORC1 do not reflect mTORC1 kinase activity. Importantly, we analyzed the levels of phospho-S6, which is mediated by an mTORC1 target p70S6K, and those of SESN2 in endometrial cancer and normal tissues using immunoblotting. The analysis indicated SESN2 and phospho-S6 levels to be significantly elevated in endometrial cancer tissues than those in normal tissues (Figure 1B and 2A). As the reviewer suggested, we have included this statement in the revised Result section [marked] (page 4 line 129-132,138-139).
- 3B: How does SESN2 regulate MKI67 mRNA expression?
Response: Tumors cell proliferation was assessed by MKI67 mRNA expression. mTORC1 is known as a key regulator of cell growth, proliferation, and survival by phosphorylation of its downstream effector molecules, S6K1 and 4E-BP1 (Fingar and Blenis 2004). Although mTORC1 is not an oncogene, as part of the PI3K/Akt signaling pathway, mTORC1 promotes cell proliferation. Recent studies have shown that SESN2 negatively regulates cell proliferation by inhibiting mTORC1 pathway (Wei, Fang et al. 2017, Luo, Zhao et al. 2018).
- The authors mentioned that SESN2 expression positively correlates with mTORC1 activity (line 31), which increases cell proliferation and is increased in endometrial cancers (Fig. 1A-1D). However, SESN2 expression was also positively correlated with cyclin-dependent kinase inhibitors CDKN1A and CDKN1B (line 171-172, Fig. 3C & D), which inhibit cell proliferation. The authors should provide an explanation.
Response: Thank you very much for this important point. The Cip/Kip family members, CDKN1A and CDKN1B, can arrest cell proliferation and initially considered as tumor suppressors (Sherr 1996). However, several studies have implicated their dual roles in cancer as tumor suppressor and oncogene (Abukhdeir and Park 2008, Llanos and Garcia-Pedrero 2016, Morris-Hanon, Furmento et al. 2017). Llanos et al. reported that CDKN1A expression is detected in proliferating head and neck squamous cell carcinomas cells and the growth inhibitory effect of CDKN1A is abrogated in these tumors (Llanos and Garcia-Pedrero 2016). We showed that knockdown of SESN2 increased mTORC1 activity and cell proliferation in HEC-1A and Ishikawa cells. However, SESN2 mRNA and protein expression was upregulated in endometrial cancer patient tissues where mTORC1 activity was still high. The results from the GEPIA database indicated statistically significant positive correlations of expression of SESN2 with CDKN1A and CDKN1B. Our data thus suggest that these correlations between SESN2 and cell cycle-associated genes are independent of the role of CDKN1A and CDKN1B as tumor suppressors in the tumor of patients with endometrial cancer. As the reviewer suggested, we have included this statement in the revised Discussion section [marked] (page 13 line 368-378).
- Figure 4C: SESN2 knockdown had little effect on E-cadherin in Ishikawa cells.
Response: Loss of E-cadherin correlates with cancer cell invasion, whereas re-expression of E-cadherin leads to inhibition of invasiveness of cancer cells. N-cadherin promotes increased cell motility and migration. Features of EMT is the upregulation of N-cadherin and parallel downregulation of E-cadherin, but this parallel regulation of cadherins is not applicable in some cases (Nakajima, Doi et al. 2004, Wheelock, Shintani et al. 2008, Araki, Shimura et al. 2011). Although knockdown of SESN2 did not significantly reduce E-cadherin protein level in Ishikawa cells, it significantly increased N-cadherin protein level. Thus, our results suggest that increased expression of N-cadherin by knockdown of SESN2 is sufficient to promote EMT and invasion in endometrial cancer cells.
- Figure 8F: While previous studies have shown that transcriptional activation of SESN2 is one of the negative feedback mechanisms for inhibiting chronic activation of mTORC1 (line 69-71), it is not clear if SESN2 also negatively regulates mTORC1 via AMPK in endometrial cancers.
Response: As the reviewer pointed out, we did not observe that SESN2 negatively regulates mTORC1 via AMPK in endometrial cancer cells. However, it has been well-known that SESN2 inhibits mTORC1 kinase activation through the activation of AMPK and phosphorylation of TSC2 (Budanov and Karin 2008, Lee, Budanov et al. 2012, Park, Park et al. 2014).
- Line 74 and 75: It was mentioned that SESN2 expression is cell-type specific. But it seems that SESN2 expression was reported to be both downregulated and upregulated in hepatocellular carcinoma.
Response: We agree and rephrased the sentence accordingly. [marked] (page 2 line 73-75).
- 321: Reference should be cited for limited studies discussing the regulation of the mTOR pathway.
Response: We are thankful for the reviewer’s suggestion. Now references are added in the Discussion section. [marked] (page 12 line 324).
Reviewer 3 Report
This is very well-designed, presented, and written manuscript investigating role of Sestrin-2 in mTORC1-dependent regulation of cellular phenotypes associated with endometrial cancer progression. The conclusion is well supported by data presented.
Author Response
Dear Reviewers,
We highly appreciate the reviewers for constructive criticisms and valuable comments. Each comment has been carefully considered point by point and responded. Specific responses to each of the comments are provided below.
This is very well-designed, presented, and written manuscript investigating role of Sestrin-2 in mTORC1-dependent regulation of cellular phenotypes associated with endometrial cancer progression. The conclusion is well supported by data presented.
Response: We thank the reviewer for the positive appraisal of our article.
Round 2
Reviewer 1 Report
The authors have provided clarifications and they did modifications in the manuscript in order to improve its quality.